# Investigation of Root Morphological Traits Using 2D-Imaging among Diverse Soybeans (*Glycine max* L.)

**DOI:** 10.3390/plants10112535

**Published:** 2021-11-21

**Authors:** Pooja Tripathi, Jamila S. Abdullah, Jaeyoung Kim, Yong-Suk Chung, Seong-Hoon Kim, Muhammad Hamayun, Yoonha Kim

**Affiliations:** 1Department of Applied Biosciences, Kyungpook National University, Daegu 41566, Korea; tripathipooza21@gmail.com (P.T.); abdullajamila@gmail.com (J.S.A.); 2Department of Plant Resources and Environment, Jeju National University, Jeju 63243, Korea; baron7798@jejunu.ac.kr (J.K.); yschung@jejunu.ac.kr (Y.-S.C.); 3National Agrobiodiversity Center, National Institute of Agricultural Sciences, RDA, Jeonju 54874, Korea; shkim0819@korea.kr; 4Department of Botany, Abdul Wali Khan University, Mardan 23200, Pakistan; hamayun@awkum.edu.pk

**Keywords:** root morphological traits, root surface area, root length, root diameter, WinRHIZO

## Abstract

Roots are the most important plant organ for absorbing essential elements, such as water and nutrients for living. To develop new climate-resilient soybean cultivars, it is essential to know the variation in root morphological traits (RMT) among diverse soybean for selecting superior root attribute genotypes. However, information on root morphological characteristics is poorly understood due to difficulty in root data collection and visualization. Thus, to overcome this problem in root research, we used a 2-dimensional (2D) root image in identifying RMT among diverse soybeans in this research. We assessed RMT in the vegetative growth stage (V2) of 372 soybean cultivars propagated in polyvinyl chloride pipes. The phenotypic investigation revealed significant variability among the 372 soybean cultivars for RMT. In particular, RMT such as the average diameter (AD), surface area (SA), link average length (LAL), and link average diameter (LAD) showed significant variability. On the contrary RMT, as with total length (TL) and link average branching angle (LABA), did not show differences. Furthermore, in the distribution analysis, normal distribution was observed for all RMT; at the same time, difference was observed in the distribution curve depending on individual RMT. Thus, based on overall RMT analysis values, the top 5% and bottom 5% ranked genotypes were selected. Furthermore, genotypes that showed most consistent for overall RMT have ranked accordingly. This ultimately helps to identify four genotypes (IT 16538, IT 199127, IT 165432, IT 165282) ranked in the highest 5%, whereas nine genotypes (IT 23305, IT 208266, IT 165208, IT 156289, IT 165405, IT 165019, IT 165839, IT 203565, IT 181034) ranked in the lowest 5% for RMT. Moreover, principal component analysis clustered cultivar 2, cultivar 160, and cultivar 274 into one group with high RMT values, and cultivar 335, cultivar 40, and cultivar 249 with low RMT values. The RMT correlation results revealed significantly positive TL and AD correlations with SA (r = 0.96) and LAD (r = 0.85), respectively. However, negative correlations (r = −0.43) were observed between TL and AD. Similarly, AD showed a negative correlation (r = −0.22) with SA. Thus, this result suggests that TL is a more vital factor than AD for determining SA compositions.

## 1. Introduction

The soybean is an affordable protein-rich source of food that also has high economic and environmental values. Therefore, soybean is one of three major crops worldwide [1]. Climate is rapidly changing, and as a result, temperature increase has led to a substantial yield decline in soybean. Thus, huge economic losses occur yearly [2]. Drought is among the severe problems that induce abiotic stresses. Therefore, it negatively impacts crop production [3,4,5]. For this reason, the yield of legumes, such as soybean, chickpeas, and lentils, has recorded lower productivity in arid and semi-arid areas due to terminal droughts in such areas [6].

Root morphology and the architecture of plants determine the productivity of plants in optimal or suboptimal environments [7]. Morphological traits, such as root length, diameter, and surface area (SA) also support not only plant growth via water and nutrient absorption, but also provide anchorage to the plant. These traits enhance nutrient availability as well by establishing a symbiotic relationship with the microflora [8,9]. Therefore, the plant root is the first plant part to ascertain drought stress from soil because insufficient water content causes soil drought [3]. Hence, understanding various root morphological traits (RMT), such as the root length, diameter, and SA, is essentially needed for crop improvement and yield increase under soil drought or nutrient deficiency conditions [10]. Except for the identified RMT, some of the root phenes, which are known as part of the root system architecture, are also involved in tolerance to drought and nutrient deficiencies [3,11,12]. Furthermore, according to Zegada-Lizarazu et al. [11], most plants consist of deep and dynamic root systems due to their abilities in enhancing water and nutrient uptake. Therefore, an investigation of root system architecture is crucial for improving drought tolerance in many crops [13,14]. Furthermore, RMT and root system architecture are important in understanding osmotic and nutrient stress. However, until now, only a few related studies have been conducted on major crops, such as maize [15], wheat [16], and rice [17]. In particular, in the case of soybean, identification of its RMT is rare. According to [18], they evaluated 49 soybean cultivars (*Glycine max* L.) to identify various RMT related characteristics, including plant shoot phenotypes, plant height, dry weight, and chlorophyll index. Except for this research, root traits have remained unidentified among a huge number of soybean cultivars because the determination of RMT is difficult due to the plasticity of root phenotypes in response to various factors, such as soil density, distribution of nutrients, and water content [19,20,21]. Furthermore, root analysis is labor-intensive work. Therefore, root research in soybean has rarely been conducted [10,22]. Researchers have developed various methods for root analyses such as semi-automated image-analysis software called SmartRoot [10,22,23]. RhizoTubes has also been used as a non-invasive image acquisition technique for plant root and shoot systems [24]. In this experiment, we used the WinRHIZO Pro image-analysis system, which is designed for root measurement in various forms. It can measure root morphology (length, area, volume, etc.), topology, and architecture and color analyses as well. It is very competitive with other semi- or fully automated methods and consists of a user-friendly computer program and a scanner for image acquisition.

It is well known that RMT is very important for the development of new climate-resilient crops. However, information on root morphological characteristics is poorly understood in soybean. Thus, to know the variation in RMT among diverse soybean for selecting superior root attribute genotypes, our research team analyzed the RMT and root system architecture of 372 cultivated soybean cultivars for the first time. This set of cultivars have been previously identified to have diverse genetics in terms of agronomic traits, so we expected to find diversity in their RMT as well. In particular, we used root scan imaging in applying high-throughput phenotype techniques to evaluate many soybean cultivars.

## 2. Materials and Methods

### 2.1. Plant Materials, Growth Conditions

Here, 372 soybean cultivars were used for evaluating RMT. The NAC at the Rural Development Administration (RDA) in South Korea donated all soybean seeds (Appendix A). Then, the seeds were sterilized with 70% ethanol (Sigma-Aldrich, MO, USA) and thoroughly rinsed with distilled water before seed propagation. Subsequently, the seeds were sown in polyvinyl chloride (PVC) pipes [6 cm (diameter) × 40 cm (height)] containing horticultural soil (Tobirang, Baekkwang Fertility, Andong, Korea). All pots were placed in a greenhouse located at the research center of the Kyungpook National University. Two seeds per pot were sown in PVC pipes. After seed germination, we selected one soybean plant for root analysis.

### 2.2. Determination of Root Phenotypes

Two-dimensional (2D) root images were used for identifying RMT. Root samples were harvested when the seedlings reached the V2 growth stage (2 trifoliate leaves). Whole soil samples were then carefully poured into a sieve, after which root samples were separated from the soil. Afterward, the roots were thoroughly washed under clean tap water and individually stored in plastic bags, which contained a small amount of water to prevent root drying. Subsequently, a scanner was used to collect the 2D root images (Epson, Expression 12,000XL, Nagano, Japan) using washed-root samples. The transparent plastic tray (30 cm long × 20 cm wide) was laid on the scanner, then clean tap water was added to the tray. Furthermore, washed-root samples were carefully moved in transparent plastic trays. The root images were captured when whole root samples floated on the water’s surface. The detailed process of root sample collection and analysis using WinRHIZO was well described in a previously published paper [22]. To minimize overlapping of roots, we carefully separated the overlapped roots using tweezers. Then, to analyze whole root images, we annotated only root areas, then analyzed these using the WinRHIZO Pro software (Regent Instruments, Inc, WinRHIZO Pro, Quebec, Canada). It takes approximately 1 min 30 s from scanning an image to analysis via the software. Other than that, the time required for root washing and adjusting it in the tray varies according to the size of the root and the amount of dirt in the root structure.

### 2.3. PCA Plot Analysis

For this study, 372 soybean cultivars were used in the PCA plot analysis. It was conducted following the PRCOMP procedure via R studio. This plot showed results of the first principal components against the second.

### 2.4. Statistical Analysis

The experimental design was randomized using three replications. All figures were also made with Microsoft Excel (2013) to evaluate the RMT among the 372 soybean cultivars. To determine statistical significance, we conducted ANOVA (SAS release 9.4; SAS, Gary, NC, USA). The correlation analysis was also conducted with R studio to investigate the relationship among RMT. Then, histograms and box plots were produced using IBM SPSS statistics 25 and SAS release 9.4; SAS.

## 3. Results

### 3.1. Seed Collection Area

One thousand (1000) soybean cultivars were donated by the National Agrobiodiversity Center (NAC) in South Korea. Among them, we randomly selected 372 soybeans for this study. Investigation of selected soybean cultivars showed that the 372 genotypes selected were collected from 18 countries. The seeds were collected from USA, China, Japan, South Korea, Russia, and North Korea, comprising 92.2%, and the other 12 countries made up the remaining 7.8% (Figure 1).

### 3.2. Variability of Root Morphological Traits

Among various RMT, we focused on length and SA. Thus, we analyzed relevant root phenotypes, such as total length (TL), the average diameter (AD), surface area (SA), link average length (LAL), link average diameter (LAD), and link average branching angle (LABA). According to the analysis of variance (ANOVA), all root traits showed significant variability among the 372 soybean cultivars selected (Table 1). Alternatively, in the statistical analysis within replication, TL and LABA did not show any difference, whereas AD, SA, LAL, and LAD showed significant variability (Table 1).

According to the distribution analysis, all RMT showed a normal distribution (Figure 2). The range of TL marked was 113 cm to 878 cm, and those more than 618 cm of TL formed part of the top 5% among 372 genotypes (Figure 2A). In contrast, AD showed ranged from 0.4 mm to 0.9 mm, and more than 0.7 mm AD was ranked in the top 5% among 372 genotypes (Figure 2B). Furthermore, the range of SA observed was from 23 cm^2^ to 144 cm^2^ per plant, and the top 5% of SA was identified as being more than 102 cm^2^ (Figure 2C). Likewise, LAL, LAD, and LABA showed similar distribution tendencies. Therefore, LAL was observed between 0.237 cm to 0.720 cm, and when the value showed over 0.507 cm, it was included in the top 5% (Figure 2D). Additionally, the distribution of LAD showed from 0.390 mm to 1.070 mm, and when LAD remarked more than 0.775 mm, the value was in the top 5% among the 372 genotypes (Figure 2E). Likewise, the LABA was recorded between 24.8° to 60.1° among the 372 genotypes, and LABA in the top 5% showed higher than 55° (Figure 2F). Contrasting root images were added in Appendix A to help to understand difference of soybean root between highest and lowest.

The list of soybean genotypes is shown in Table 2. As shown, each root trait was arranged sequentially, then only 5% of genotypes from the highest and lowest were selected. Consequently, 73 (highest) and 72 (lowest) genotypes were included in the selected 5% at more than one time (data not shown). Among them, most genotypes ranked 5% below twice. Thus, we selected soybean genotypes, which ranked in 5% more than thrice (Table 2). According to Table 2, a distinguished tendency was observed among selected soybean genotypes in the highest and lowest 5%. In the highest 5%, four genotypes ranked 5% more than thrice among the six RMT (Table 2). However, among the four genotypes, IT 165308 and IT 199127 ranked 5% in TL, SA, and LABA, whereas the other two genotypes (IT 165432 and IT 165282) recorded 5% in AD, LAL, and LAD (Table 2). Alternatively, in the lowest 5%, nine genotypes ranked 5% more than thrice. Among these nine genotypes, IT 23305 and IT 208266 recorded 5% more than four times. Commonly, SA and LABA recorded 5% in both genotypes (Table 2). Among the other seven genotypes, four genotypes (IT 165208, IT 156289, IT 165405, and IT 165019) commonly ranked 5% in TL and SA, whereas three genotypes (IT 165839, IT 203565, and IT 181034) equally ranked 5% in the AD, LAL, and LAD groups (Table 2).

Based on the seed collection region, we compared the root morphological traits of selected cultivars (Figure 3). Most seed collection regions showed high variations among the six RMT. In particular, TL, SA, and LAD showed higher variability than AD, LAL, and LABA (Figure 3). However, among the seed collection regions, PRK showed the highest variability in TL and SA, even though only 2.69% of seeds were collected from that area. In the case of LABA, most of the countries showed a low variation (Figure 3F).

We conducted a principal component analysis (PCA) based on all phenotypic data obtained, and generated a biplot to inspect the possibility of clustering of the cultivars (Figure 4A). The biplot shows the loading of each variable (arrows) and scores of each cultivar (numbers). Additionally, the angles between the arrows show their approximate correlations. The numbers close together in the plot are the observations with similar scores on the PCA components. The biplot illustrates the clustering of these cultivars. Therefore, cultivar 2—Crawford (IT 21595), cultivar 160—Burlison (IT 165023), and cultivar 274—Jiu nong 1 (IT 165839) were included in a cluster and ranked among the highly ranked cultivars based on mean values. Likewise, cultivar 335—Zaredo F-267 (IT 208266), cultivar 40—Hood (IT 22314), and cultivar 249—PI467332 (IT 165297) formed another cluster, and they were included in the low-ranked cultivars (Figure 4A). These clusters were separated from other cultivars, which showed that the values for various root traits for those cultivars were different from those of other cultivars. In the case of PCA, it creates principal components which are the linear combination of diverse factors. Thus, it is difficult to interpret complex relationships among the factors. Therefore, we conducted t-SNE because it is an effective non-linear dimensionality reduction algorithm (Figure 4B). The t-SNE data were adjusted by 2 dimensionalities, and we confirmed three major distinguished clusters according to RMT (Figure 4B).

### 3.3. Correlation

Using the six root traits, we conducted a correlation test. According to the results, a strongly distinguished tendency was observed among RMT. Root length-related traits, such as TL, and SA showed significantly positive correlations (r = 0.96) than others (Figure 5). Alternatively, AD showed the highest Pearson’s correlation coefficient (r = 0.85) with LAD compared to other traits (Figure 5). However, in the case of TL, significant negative correlations were observed in AD (r = −0.43), LAL (r = −0.30), and LAD (r = −0.39), except for SA and LABA (Figure 5). In contrast, AD showed significantly positive correlations with LAL (r = 0.44) and LAD (r = 0.85); however, AD revealed negative correlations with TL (r = −0.43), SA (r = −0.22), and LABA (r = −0.051) (Figure 5). To decrease the false discovery rate, we conducted a Benjamini–Hochberg statistical test with a 20% false discovery rate (Table 3). The highest p-value that was found to be also smaller than the critical value was AD*LABA, therefore all the values above it can be considered significant, even if those p-values are lower than the critical values with the B-H correction (Table 3).

## 4. Discussion

The association of root and shoot traits, including its contribution to plant productivity, has been recognized in soybean [18]. According to [18], high-yielding soybean cultivars not only have faster root growth but also assimilate more carbohydrates into roots than low-yielding cultivars. Hence, highly improved root traits, such as length, SA, and tips, contribute to increased productivity. For this reason, many researchers believed that genetically improved RMT is attributed to a high rate of yield increase in soybean [25]. Furthermore, the relationship between the genetic variation of root traits in relation to the availability of water and nutrient absorption has also been documented in crops, such as maize (*Zea mays* L.) [26,27], spring wheat (*Triticum* spp.), and faba bean (*Vicia faba* L.) [28,29,30]. These crops showed wide variations in their roots. It has been reported that plants with broad root biomass would readily have increased water and nutrient uptake.

From the viewpoint of legume breeding to improve RMT, analysis of quantitative trait loci (QTL) is one of the powerful tools [31,32]. Thus, to identify QTLs involved in phenome characteristics, genotype and phenotype data are necessary [33]. In the case of shoot phenotypes, those are the features that are exposed to the atmosphere, and can be collected more easily than those of the root phenotypes because roots exist underground. For this reason, many researchers ignore this aspect of plant research because root research is time-consuming and labor-intensive [10]. Nevertheless, root research is important in understanding crop production processes in addition to resistance to abiotic stress conditions, such as drought and salinity. Sadly, only a few relevant studies have been conducted, due to the complexity of root harvesting in situ, and root trait evaluation [7,34]. Thus, in this study, we used 372 soybean genotypes and analyzed RMT using 2D images to overcome the limitation of root research in this study.

According to the ANOVA results, all RMT showed considerable variability among the 372 genotypes. In particular, AD, SA, LAL, and LAD showed significant differences between the replications; however, TL and LABA showed no significant difference in the replications. This result, therefore, suggested that each soybean genotype had less variability in TL and LABA than AD, SA, LAL, and LAD. Therefore, root diameter, lateral length, and SA characteristics showed wide variation. However, root length and root angle showed shallow variability among diverse soybean cultivars understudied. Similarity results were reported by Fried et al. (2018). According to these reports, even though length-related parameters showed significant differences, the average root diameter did not show any difference among the 49 soybean accessions.

The major strength of our research is the huge number of soybean genotypes that were used for identifying RMT by the 2D images. Recently, many plant studies have tried to measure phenotypes, such as plant height [35], and leaf area [36] using image data. Historically, most of these plant phenotypes, such as height, length, width, and area were measured manually. Thus, conventional methods had limitations of throughput. Alternatively, image-based phenotyping can increase the scale, throughput, and speed of data analysis [37]. Therefore, in this research, we analyzed RMT using the WinRHIZO software. The WinRHIZO software was specifically developed for measuring washed roots from various plants [38]. According to the manufacturer’s introduction, WinRHIZO technology can provide convenient and fast data collection because this system does not need calibration each time after the optical setup. Thus, this root analysis system has been used in various crops, such as bermudagrass [38], soybean [10], winter wheat [39], and maize [40]. Hence, the WinRHIZO software has been used in this study as well as various crops for a long time. Furthermore, it was continuously updated to increase accuracy. Therefore, the WinRHIZO software provided an analysis of root morphology (total length, average diameter, total area, tip, forks) and root architecture (branching angle). Among them, we focused on root morphology. Thus, we collected six morphological traits (TL, AD, SA, LAL, LAD, and LABA) and used WinRHIZO software for analysis. Based on the analyzed root data, we conducted distribution analyses, then we confirmed that all root traits showed a normal distribution (Figure 2). Subsequently, we screened extremely contrasting genotypes of each root trait because all root traits showed a normal distribution. To conduct this screening, we lined up the 372 genotypes by every single root trait, then selected the highest and lowest ranking 5% of these genotypes. Based on the six RMT studied, the 73 (highest) and 72 (lowest) genotypes were recorded in the 5% ranking more than once. Furthermore, although many genotypes were included in the highest and lowest 5%, only a few genotypes remained when we enhanced the criterion to those that met more than 5% three times. These results suggested that multiple RMT should be considered for the selection of reinforced soybean root traits because many one-off emergences had been investigated in RMT.

If plants possess long root lengths or increased SA, the possibility of water and nutrient absorption increases [3]. Therefore, the length of root segments influences root lengths, whereas, root length affects root SA in addition to root width [3,22]. From the viewpoint of root biomass, therefore, root length and diameter are considered to be the most important traits. Thus, our study investigated the minimization of the gap that exists between root length and root diameter. TL showed a significantly positive correlation with SA (r = 0.96). Furthermore, TL also showed a significant correlation with AD (r = −0.43). Thus, although, TL showed a significant correlation with both traits, TL showed stronger correlations with SA when compared to the Pearson’s value. Therefore, these results reveal that SA was more affected by TL than AD. Similarity results were also reported in a previous study [18]. According to them, even though the total root length showed a significant correlation with total root SA (r = 0.93), total root length did not show a correlation with average root diameter [18]. Alternatively, TL was a key factor of SA; however, LAD was a major component of AD according to our result (Figure 5). Therefore, these results reveal that TL mainly affected SA, but AD influenced LAD. Thus, we hypothesized that in fact, TL and AD were distinguished as RMT. Therefore, both traits possessed incompatible root phenotypes during the selection of elite germplasms. Additionally, in comparison among seed collection regions, insignificant differences among seed collection areas were observed (Figure 4). A similar result was reported [18]. In their study, they conducted an analysis of soybean root traits among 49 genotypes, which included diverse maturity groups (MG) (IV, V, VI, VII, and VIII), and they reported that maturity groups did not affect RMT.

Classification systems of MG in soybean have also been developed in North America, and 13 MGs that were classified ranged from MG 000 to MG X [41]. This classification was because soybean is a typical short-day plant. Therefore, the photoperiod and temperature were the most important factors for flowering. The study showed that soybean had a narrow range of latitudes depending on the flowering season [42]. In our research, the 372 soybean cultivars collected were not only from various countries but also from several latitudes. Our results revealed that regional differences were not major factors related to RMT, especially, in TL and SA. Perhaps, we anticipated that those results were caused by several possibilities. First, we propose that other environmental factors, such as soil physical properties, soil chemical properties, soil moisture and soil nutrient content, are more major factors for root morphogenesis than regional differences. Second, the unique properties of soybean cultivars involving their ability in water and nutrient absorption is also a component for the formation of RMT. Therefore, to confirm those hypotheses, additional studies are required using specific soybean germplasms.

## 5. Conclusions

In this study, we analyzed the RMT involved in root length and roots from diverse soybean genotypes using 2D image technology in 372 soybeans for the first time. According to ANOVA, most root traits showed a normal distribution. Thus, we selected elite cultivars based on the highest and lowest 5% values from each root trait. As reported, 73 and 72 genotypes were listed on the 5% highest and 5% lowest rankings more than once, respectively. When we enhanced the criteria that genotype should rank in 5% more than three times, four genotypes (IT 16538, IT 199127, IT 165432, and IT 165282), and nine genotypes (IT 23305, IT 208266, IT 165208, IT 156289, IT 165405, IT 165019, IT 165839, IT 203565, and IT 181034) were selected in the highest and lowest, respectively. The selected genotypes also showed common patterns. If soybean showed large SA, long TL was accompanied. However, if soybean had wide AD, then wide LAD and long LAL were observed. Similar results were investigated using the correlation test. As revealed, TL showed a significantly positive correlation with SA (r = 0.96), and AD revealed significantly positive correlations with LAD (r = 0.85). However, negative correlations were observed between TL and AD (r = −0.43). Consequently, our results are summarized below. First, root biomass principally affects root length. Second, root segment diameter influences root diameter. Third, although both root traits affect the SA, including length and diameter, root length is a more key factor than the diameter in the increase of root SA. Therefore, TL should be preferentially considered during the selection of soybean’s large root biomass.

## Figures and Tables

**Figure 1 plants-10-02535-f001:**
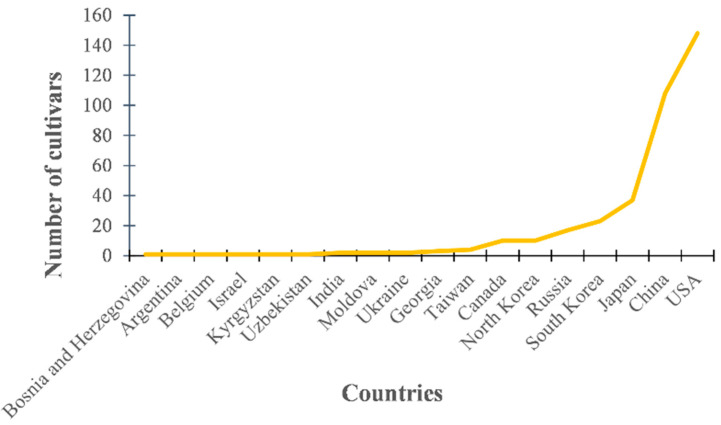
Variation in seed collection regions among 372 soybean cultivars selected for this study.

**Figure 2 plants-10-02535-f002:**
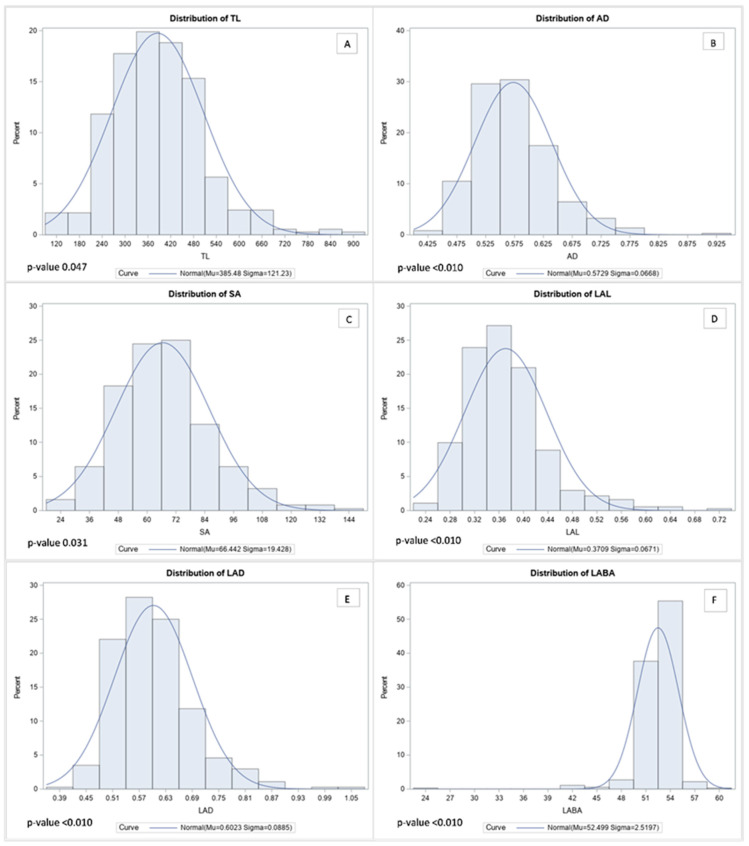
Histogram of normal distribution curves for root morphological traits. The Kolmogorov–Smirnov test was used for testing the normality of the distribution. In the figure, each abbreviation indicated total length (TL), average diameter (AD), surface area (SA), link average length (LAL), link average diameter (LAD), and link average branching angle (LABA), respectively. In the figure, each capital letters indicate distribution of root morphological traits such as TL (**A**), AD (**B**), SA (**C**), LAL (**D**), LAD (**E**), and LABA (**F**), respectively.

**Figure 3 plants-10-02535-f003:**
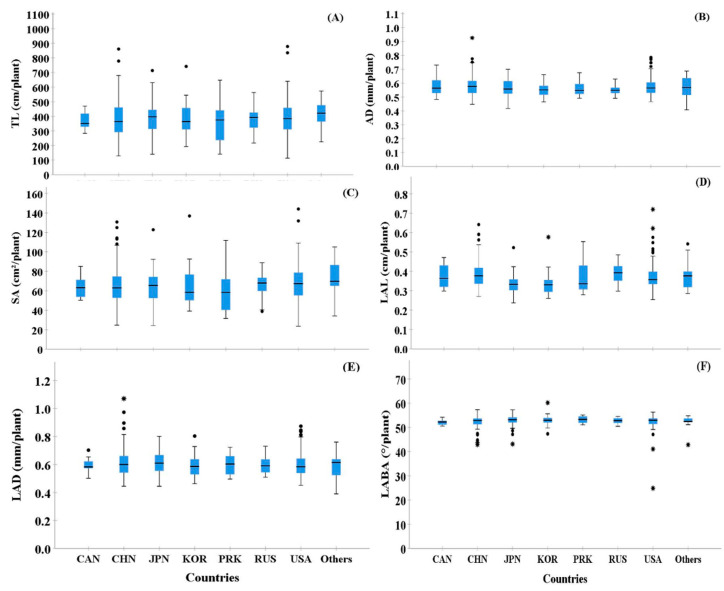
Box plots showing the shape of the distribution, central value, and variability of root traits from the 372 cultivars examined based on the eight-country groups, which were separated by collection regions. In the figure, ° and * indicate high and extremely high values in the observations, respectively, and are called outliers in the data. In the figure, **A**–**E** indicate following: TL (**A**), AD (**B**), SA (**C**), LAL (**D**), LAD (**E**), and LABA (**F**).

**Figure 4 plants-10-02535-f004:**
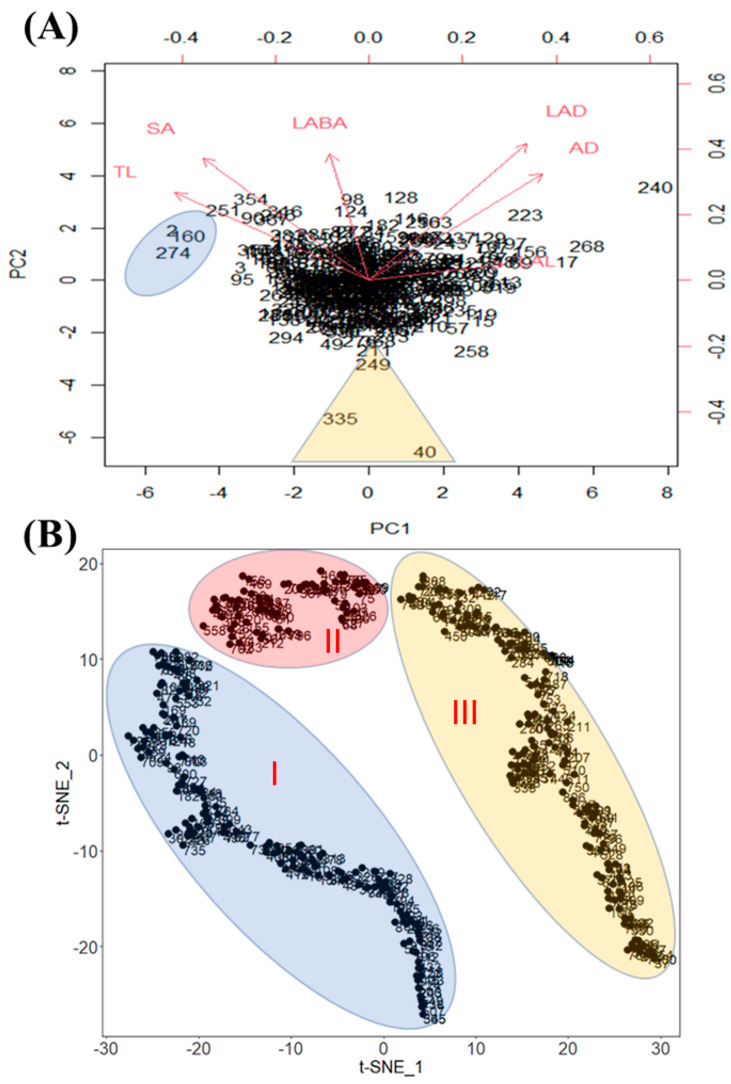
Principal Component Analysis biplot that separated the soybean cultivars into clusters based on root traits. In the figure, (**A**) indicated that PC1 accounted for 46.87% of the total variation, and PC2 accounted for 23.89% of the total variation. However, (**B**) was analyzed by the t-SNE method and each cluster indicates genotypes that have similar root traits.

**Figure 5 plants-10-02535-f005:**
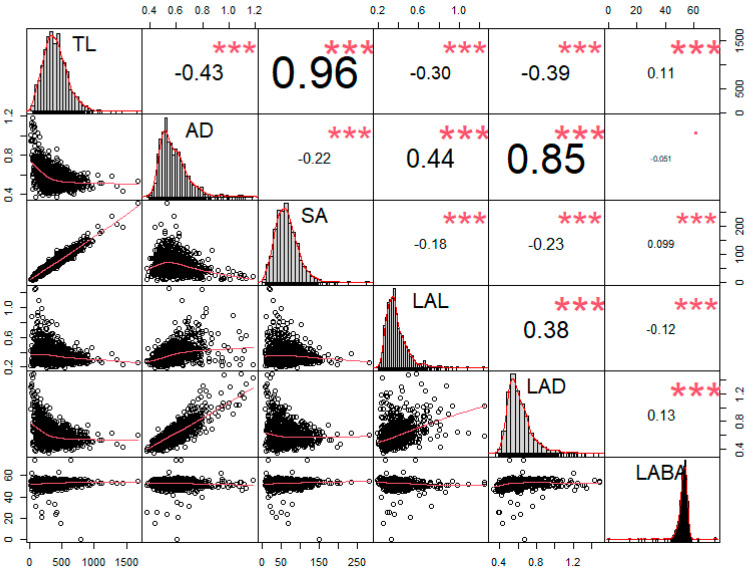
Correlation analysis between the six root parameters. Asterisks (*) indicate statistically significant scores, where (*) indicates significance at 0.05 level, (**) at 0.01, and (***) at 0.001 level.

**Table 1 plants-10-02535-t001:** Analysis of variance (ANOVA) results from root morphological traits.

**Trait**	Source	DF	Sum of Squares	Mean Square	F Value	Pr > F
**TL**	variety	371	16,319,131	43,986.88	1.29	0.0022
	rep	2	50,091.84	25,045.92	0.73	0.481
**AD**	variety	371	4.9579692	0.0133638	1.73	<0.0001
	rep	2	2.7102099	1.3551049	175.8	<0.0001
**SA**	variety	371	418,755.69	1128.7215	1.33	0.0007
	rep	2	27,824.935	13912.468	16.34	<0.0001
**LAL**	variety	371	4.9751331	0.0134101	1.23	0.0107
	rep	2	1.3963546	0.6981773	63.85	<0.0001
**LAD**	variety	371	8.7093611	0.0234754	1.31	0.0012
	rep	2	2.7739838	1.3869919	77.36	<0.0001
**LABA**	variety	371	7059.1998	19.027493	1.5	<0.0001
	rep	2	36.150078	18.075039	1.43	0.2401

TL: Total length, AD: average diameter, SA: surface area, LAL: link average length, LAD: link average diameter, LABA: link average branching angle.

**Table 2 plants-10-02535-t002:** Information on soybean genotypes included in the 5% highest and lowest groups that occurred more than three times among the six morphological traits. The star mark in the table means that the genotype was included in 5% of each side. In the table, each abbreviation indicated total length (TL), average diameter (AD), surface area (SA), link average length (LAL), link average diameter (LAD), and link average branching angle (LABA), respectively.

Genotype		Traits					
		TL	SA	AD	LAL	LAD	LABA
Highest 5%	IT 165308	*	*				*
	IT 199127	*	*				*
	IT 165432			*	*	*	
	IT 165282			*	*	*	
Lowest 5%	IT 23305	*	*		*		*
	IT 208266		*	*		*	*
	IT 165208	*	*			*	
	IT 156289	*	*				*
	IT 165405	*	*				*
	IT 165019	*	*		*		
	IT 165839			*	*	*	
	IT 203565			*	*	*	
	IT 181034			*	*	*	

**Table 3 plants-10-02535-t003:** Benjamini–Hochberg test for correlated variables.

Correlated Variables	*p*-Value	Rank	(i/m)/Q
TL×SA	0.000000	1	0.0033333
TL×AD	0.0000000	2	0.0066667
TL×LAL	0.0000000	3	0.0100000
TL×LAD	0.0000000	4	0.0133333
SA×AD	0.0000000	5	0.0166667
SA×LAL	0.0000000	6	0.0200000
LAL×LAD	0.0000000	7	0.0233333
AD×LAL	0.0000000	8	0.0266667
AD×LAD	0.0000000	9	0.0300000
TL×LABA	0.0002000	10	0.0333333
SA×LAD	0.0002000	11	0.0366667
SA×LABA	0.0002000	12	0.0400000
LAD×LABA	0.0016000	13	0.0433333
LAL×LABA	0.0198000	14	0.0466667
AD×LABA	0.2604000	15	0.0500000

In the table, i = the individual *p*-value’s rank, m = total number of tests, Q = the false discovery rate.

## Data Availability

Not applicable.

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
