# Peer review of "Investigation of Root Morphological Traits Using 2D-Imaging among Diverse Soybeans (Glycine max L.)"

_plants, 2021, doi:10.3390/plants10112535_

Round 1

Reviewer 1 Report

First of all, I would like to express my gratitude to the authors for correcting most of the comments that I had on the first draft of the article. The authors added the Kolmogorov-Smirnov normality test, t-SNE graph, images of contrasting roots in supply.

I still have a few minor comments related to improving the visual perception of the article results.

Figure 4A:

Genotypes 274, 160, 2 and 335, 249, 40 are not obvious clusters.

The figure has not changed and is not informative.

Figure 4B

The authors added t-SNE. Clusters can be distinguished better, but the numbers merge again and are not cheats. Figures 4F and 4B need to be improved.

By what principle are genotypes designated either together or separately? For example, in the abstract: “cultivar2, cultivar 160, and cultivar 274”. Everything needs to be done uniformly.

On line 183, there seems to be a typo in the genotype number 249, because for the rest of the numbers (IT XXXXXX) are the same:

183: “208266), cultivars 40 - Hood (IT 22314), and cultivars 249 - PI467332 (IT 165432) formed”

Table 2: the heading shows the abbreviation “TRL”, while the description and the rest of the text have TL (total length).

Figure 3.

171-172: “In the figure, ° and * indicate high and extremely high values ​​in the observations, respectively, and are called outliers in the data.”

The drawing needs to be improved. It is necessary to increase the resolution and clarity. Asterisks are poorly visible.

398: Supplementary Materials: Table S1: List of soybean genoytpes.

Typo.

Figure S1.

It is necessary for each image to indicate the values ​​of the traits: TL: Total length, AD: average diameter, SA: surface area, LAL: link average length, LAD: link av-112 erage diameter, LABA: link average branching angle.

Author Response

Author’s pointwise response to the review report (Reviewer 1)

First of all, I would like to express my gratitude to the authors for correcting most of the comments that I had on the first draft of the article. The authors added the Kolmogorov-Smirnov normality test, t-SNE graph, images of contrasting roots in supply.

I still have a few minor comments related to improving the visual perception of the article results.

Point 1: Figure 4A: Genotypes 274, 160, 2 and 335, 249, 40 are not obvious clusters. The figure has not changed and is not informative.

Response 1: We appreciate the reviewer for his valuable comments and suggestions. The authors agree that the visual look congested and it's not having obvious clusters. However, we have tried all the possible combination in the setting but as the number of genotypes are more it's impossible to get further separation in Figure 4A. But we improved the image by showing mentioned genotypes highlighted with a cluster. We would like to apologize for this inconvenience, despite the high interest we are unable to improve further separation. Secondly, we have clearly explained the figures in the result section with the details about these genotypes. So, it may not cause any confusion to the reader.

Point 2: Figure 4B The authors added t-SNE. Clusters can be distinguished better, but the numbers merge again and are not cheats. Figures 4F and 4B need to be improved.

Response 2: We appreciate the reviewer for his valuable comments and suggestions. As explained above further separation is not possible with the existing setting. We would like to apologize for this inconvenience. But we have shown three major clusters in t-SNE Figure 4B graph. Hope it solves the concern and is clearer for the reader.

Point 3: By what principle are genotypes designated either together or separately? For example, in the abstract: “cultivar2, cultivar 160, and cultivar 274”. Everything needs to be done uniformly.

Response 3: Thank you for commenting on it. We would like to say that these genotypes are separated based on the similar/diverse (high or low) root traits values. Similar we have mentioned in the abstract with the track change.

Point 4: On line 183, there seems to be a typo in the genotype number 249, because for the rest of the numbers (IT XXXXXX) are the same: 183: “208266), cultivars 40 - Hood (IT 22314), and cultivars 249 - PI467332 (IT 165432) formed”

Response 4: Thank you for pointing this. We apologies for the typo, it’s a typo error. Now it is corrected in the main text with track change.

Point 5: Table 2: the heading shows the abbreviation “TRL”, while the description and the rest of the text have TL (total length).

Response 5: Sorry for the inconvenience we have changed it to TL.

Point 6: Figure 3. 171-172: “In the figure, ° and * indicate high and extremely high values ​​in the observations, respectively, and are called outliers in the data.” The drawing needs to be improved. It is necessary to increase the resolution and clarity. Asterisks are poorly visible.

Response 6: Sorry for the inconvenience we have changed Figure 3 and improved the resolution according to the suggestion

 Point 7: 398: Supplementary Materials: Table S1: List of soybean genoytpes.

Typo.

Response 7: Thank you for pointing this. We apologies it’s a typo now it is corrected.

Point 8: Figure S1. It is necessary for each image to indicate the values ​​of the traits: TL: Total length, AD: average diameter, SA: surface area, LAL: link average length, LAD: link av-112 erage diameter, LABA: link average branching angle.

Response 7: Thank you for the suggestion. We apologies for the inconvenience, it is corrected now.

Reviewer 2 Report

Dear Authors

The subject of your paper is of great interest for the biologists to better understand the relationship between roots and growth of a plant.

Nevertheless there are lots of mistakes and problems in your paper which should be addressed to improve the content and the results.

  • The abstract needs to be improved to better correspond to the title of your paper
  • In terms of structure of the paper, the "results" part must be after the "material and methods" !
  • You never explain why you decided to use WinRHIZO rather than another commercial software like Smartroot for example ?
  • You mention that the data come from several areas around the world but you never explain the differences in terms of soil, climate ... which could explain the different in the results !
  • The AD parameter is not the most interesting. Some research have showed that we can determine the root diameter every mm, for example. Indeed the average diameter is highly depending on the genotype and the susbtract/nutrients.
  • The references are for some of them rather old (before 2010) whereas a lot of researches have been done after. You never mention for example the works of Lobet or Jeudy who have worked on a comparison of different software (Lobet) or on the conception of Rhizotubes (Jeudy) to better see the roots during all the life of the plants.
  • Finally you never give some information on the processing time !

Author Response

Author’s pointwise response to the review report (Reviewer 2)

The subject of your paper is of great interest for the biologists to better understand the relationship between roots and growth of a plant.

Point 1: Nevertheless there are lots of mistakes and problems in your paper which should be addressed to improve the content and the results.

Response 1: We would like to thank the worthy reviewer for their time and constructive comments and suggestions to improve our manuscript. We have revised the MS, addressed all the comments, and made the suggested corrections. We hope that the revised manuscript would be suitable for publication now. All the corrections have been made with track change. The manuscript has been proofread and edited by native English language speakers.

Point 2: The abstract needs to be improved to better correspond to the title of your paper

Response 2: Thank you for the valuable suggestion we have modified the abstract.

Point 3: In terms of structure of the paper, the "results" part must be after the "material and methods" !

Response 3: Thank you for the valuable suggestion we have modified the structure.

Point 4: You never explain why you decided to use WinRHIZO rather than another commercial software like Smartroot for example ?

Response 4: Thank you for commenting on it. But we would like to draw the worthy reviewer's attention that we have clearly explained about the WinRHIZO software. In the discussion section. Line -277-286. Further, now we have mentioned why we used it. Hope it clarifies the concern.

“The WinRHIZO software was specifically developed for measuring washed roots from various plants [33]. According to the manufacturer’s introduction, the WinRHIZO technology can provide convenient and fast data collection because this system does not need calibration each time after the optical setup. Thus, this root analysis system has been used in various crops, such as bermudagrass [33], soybean [10], winter wheat [34], and maize [35]. Hence, the WinRHIZO software has been used in various crops for a long time. Furthermore, it was continuously updated to increase accuracy. Therefore, the WinRHIZO software provided an analysis of root morphology (total length, average diameter, total area, tip, forks) and root architecture (branching angle)”.

Point 5: You mention that the data come from several areas around the world but you never explain the differences in terms of soil, climate ... which could explain the different in the results !

Response 5: Sorry for the inconvenience we have mentioned that cultivars are collected or originated from several areas around the world (“the 372 genotypes selected were collected from eighteen countries”) but we did not collect data around the world.

Point 6: The AD parameter is not the most interesting. Some research have showed that we can determine the root diameter every mm, for example. Indeed the average diameter is highly depending on the genotype and the susbtract/nutrients.

Response 6: Thank you for commenting on this but we have not said that AD is most interesting. We agree that research has shown that they can determine the root diameter every mm and highly depending on the genotype and the substrate/nutrients. It would be really helpful if the reviewer pointed to the line or page where we have mentioned the above point. We can address and correct them accordingly. 

Point 7: The references are for some of them rather old (before 2010) whereas a lot of researches have been done after. You never mention for example the works of Lobet or Jeudy who have worked on a comparison of different software (Lobet) or on the conception of Rhizotubes (Jeudy) to better see the roots during all the life of the plants.

Response 7: Sorry for the inconvenience we have updated and modified the references. Our objective in this study is to evaluate the root morphological traits using 2D-imaging among diverse soybeans (Glycine max L.). Secondly, we have mentioned in the discussion section why we used this method for root trait evaluation and also provided additional information in the introduction with track change. Moreover, showing a comparison of different software may inflate our objective of study.

Point 8: Finally you never give some information on the processing time !

Response 8: Thank you for your suggestion, we have added the processing time information on page 4, line 149-152

“It takes approximately 1 minute 30 seconds from scanning an image to analysis via the software. Other than that, the time required for root washing and adjusting it in the tray varies according to the size of the root and the amount of dirt in the root structure”. 

Reviewer 3 Report

The topic is relevant to the profile of Plants and also hold intriguing findings about differences in root anatomy.

The major concern is that there are methods with the help of which root characteristics can be studied,  in vivo, such as electrical impedance measurement. It would worth including aspects into the Introduction part, why this analysis provides more information,  than other methods, especially because of the non-invasiveness. In contrast this imaging is very informative, but also very laboruous.

lines 83-86 'For this reason, our research team analyzed the root morphological traits and root system architecture of soybean using 372 cultivated soybean cultivars for the first time. In  particular, we used the root scan imaging in applying high-throughput phenotype techniques to evaluate the huge number of soybean cultivars investigated. '

Please explain the reason for conducting this research better., also indicate briefly the main findings of the experiment.

line 91 specimen or cultivars? It is definitely not species, this work is about one species: soy.

What was the reason for choosing specifically those cultivars, that were included into the experiment?

line 303 'According to our results, the definite gap that exists between root 
 length and root diameter was investigated.' 

Please rephrase this sentence, since it is not understandable in this form.

Author Response

Author’s pointwise response to the review report (Reviewer 3)

The topic is relevant to the profile of Plants and also hold intriguing findings about differences in root anatomy.

We would like to thank the worthy reviewer for their time and constructive comments and suggestions to improve our manuscript. We have revised the MS, addressed all the comments, and made the suggested corrections. We hope that the revised manuscript would be suitable for publication now. All the corrections have been made with track change. The manuscript has been proofread and edited by native English language speakers

Point 1: The major concern is that there are methods with the help of which root characteristics can be studied,  in vivo, such as electrical impedance measurement. It would worth including aspects into the Introduction part, why this analysis provides more information,  than other methods, especially because of the non-invasiveness. In contrast this imaging is very informative, but also very laboruous.

Response 1: Thank you for commenting on it. We have added the suggested information about the method used in the introduction. Previously, we have mentioned why we used WinRHIZO software analysis in the discussion part. Please refer to page 1, Line -277-286.

“The WinRHIZO software was specifically developed for measuring washed roots from various plants [33]. According to the manufacturer’s introduction, the WinRHIZO technology can provide convenient and fast data collection because this system does not need calibration each time after the optical setup. Thus, this root analysis system has been used in various crops, such as bermudagrass [33], soybean [10], winter wheat [34], and maize [35]. Hence, the WinRHIZO software has been used in various crops for a long time. Furthermore, it was continuously updated to increase accuracy. Therefore, the WinRHIZO software provided an analysis of root morphology (total length, average diameter, total area, tip, forks) and root architecture (branching angle)”.

Point 2: lines 83-86 'For this reason, our research team analyzed the root morphological traits and root system architecture of soybean using 372 cultivated soybean cultivars for the first time. In  particular, we used the root scan imaging in applying high-throughput phenotype techniques to evaluate the huge number of soybean cultivars investigated. '

Please explain the reason for conducting this research better., also indicate briefly the main findings of the experiment.

Response 2: Thank you for your comments and suggestions. We have added an explanation for why we conducted this research. All the changes mark with track change.

Point 3: line 91 specimen or cultivars? It is definitely not species, this work is about one species: soy.

Response 3: Sorry for the inconvenience we have corrected the information.

Point 4: What was the reason for choosing specifically those cultivars, that were included into the experiment?

Response 4: These are genetically diverse soybean cultivars that are widely used in the breeding program as well as they have interesting and contrasting agronomics traits. Thus we wanted to investigate root system architecture (RSA) traits among these cultivars to give insight into the root traits of these cultivars. In addition, it is shown that better RSA also influences and increases the crop yield as well as can play a major role in plant survival. Now we have provided this information.

Point 5: line 303 'According to our results, the definite gap that exists between root 
 length and root diameter was investigated.' Please rephrase this sentence, since it is not understandable in this form.

Response 5: Thank you for the suggestion, we have modified the sentence as below.

“Thus, in our study to minimize a gap exists between root length and root diameter was investigated”

Round 2

Reviewer 2 Report

Dear Authors

I would like to thank for your corrections which allow to better understand your research and the results obtained.

As you mention that you use 2D-images, you don't show sufficient of them. As previously said, winRhizo is to my opinion not the best software to use in this case.

However, the main remarks done in my previous review have been took into account.

This manuscript is a resubmission of an earlier submission. The following is a list of the peer review reports and author responses from that submission.

Round 1

Reviewer 1 Report

Pooja Tripathi and et al used high-throughput 2D imaging phenotyping techniques to assess the morphological traits of soybean roots. The plants were grown in PVC pots. Upon reaching a certain stage of growth (V2), root samples were collected, cleared of soil and scanned.

The resulting images were analyzed using WinRHIZO pro software to extract 6 morphological features of the roots: TL, total length; AD, average diameter; SA, surface area; LAL, link average length; LAD, link average diameter; LABA, link average branching angle.

Analysis of variance (ANOVA) showed that all traits had significant variability among 372 selected soybean varieties, and that root traits were normally distributed. PCA and correlation analysis performed.

There are the following comments and questions to improve this work:

Figure 1 is better to be converted into a table or bar chart, because a large number of segments with a value less than 2% are visually poorly perceived.

The authors indicate that the traits they obtained have a normal distribution, referring to the graphs in Figure 2 and the ANOVA results. This is not proof that the distribution is normal. It is necessary to perform a statistical test and indicate the p-value for the hypothesis of normality.

The distributions in Figure 5 and Figure 2 look slightly different. In Figure 5, the distributions have a long right “tail”, while in Figure 2 there are no such “tails”. What is the reason for this?

When carrying out the correlation analysis (Figure 5), many comparisons were made ((6x5) / 2 = 15), therefore, when calculating the significance level, it makes sense to correct for multiple comparisons. For example, dividing the current significance level by 15 or using the Benjamini Hochberg statistical test.

It would be interesting to group genotypes into drought-tolerant and non-drought tolerant and compare how the root parameters for these two groups will differ. This will make the article better.

An example of analyzed images of roots is not given. In the article, it is necessary to show several images (3-5) of the roots of contrasting morphotypes: with large and small values of the parameters or from different parts of the scattering cloud in Fig. 4. On each of the images it is necessary to indicate the values of the main parameters (TL, AD, SA, LAL, LAD, LABA).

In addition to the PCA (Figure 4), it would be nice to construct a non-metric scaling of the samples, for example tSNE or MDS.

Figure 4 shows two diagrams, but this is not indicated in the caption. It is not clear what the difference is. The styles of the drawings are different. It is necessary to arrange in the same style. The numbers merge in the center of the graph. Better to fix it.

Lines 349-350: Two seeds per pot were sown in PVC pipes. After seed germination, we selected one soybean plant for root analysis.

Was the plant chosen at random or was there some kind of criterion?

Lines 353-354: Root samples were harvested when the seedlings reached the V2 growth stage (2 trifoli- 353 ate leaves).

Why was this growth stage chosen?

Line 192: (Figure 4A). The biplot shows the loading of each variable (arrows) and scores of each

Throughout the text, the letter designations of individual figures are lowercase, but here is an uppercase “A”.

Lines 401-402 not referenced

Supplementary Materials: The following is available online at http: // ???, Table S1: List of soybean genotypes.

Author Response

Reviewer 1 comment

There are the following comments and questions to improve this work:

Figure 1 is better to be converted into a table or bar chart because a large number of segments with a value less than 2% are visually poorly perceived.

Answer: We changed the pie chart to a line graph for better visibility of individual data.

The authors indicate that the traits they obtained have a normal distribution, referring to the graphs in Figure 2 and the ANOVA results. This is not proof that the distribution is normal. It is necessary to perform a statistical test and indicate the p-value for the hypothesis of normality.

Answer: We have changed figure 2 accordingly. P-values for testing normality have been added from The Kolmogorov–Smirnov test

The distributions in Figure 5 and Figure 2 look slightly different. In Figure 5, the distributions have a long right “tail”, while in Figure 2 there are no such “tails”. What is the reason for this?

Answer: The distribution graphs in Figure 2 also have a tail on the right but it has been merged with the x-axis of the graph. That is why it isn’t visible distinctly.

When carrying out the correlation analysis (Figure 5), many comparisons were made ((6x5) / 2 = 15), therefore, when calculating the significance level, it makes sense to correct for multiple comparisons. For example, dividing the current significance level by 15 or using the Benjamini Hochberg statistical test.

Answer: Thank you for the comment. We have added a table correcting with the B-H procedure for the correlation analysis.

It would be interesting to group genotypes into drought-tolerant and non-drought tolerant and compare how the root parameters for these two groups will differ. This will make the article better.

Answer: Thank you for your suggestion. It definitely would make this article better. We received the soybean genotypes from the seed bank of Korea and unfortunately, we don’t have the information to group them into drought-tolerant or susceptible as many of them might not have been explored yet.

An example of analyzed images of roots is not given. In the article, it is necessary to show several images (3-5) of the roots of contrasting morphotypes: with large and small values of the parameters or from different parts of the scattering cloud in Fig. 4. On each of the images, it is necessary to indicate the values of the main parameters (TL, AD, SA, LAL, LAD, LABA).

Answer: We have added an image showing contrasting genotypes as a supplementary file.

In addition to the PCA (Figure 4), it would be nice to construct a non-metric scaling of the samples, for example, tSNE or MDS. Figure 4 shows two diagrams, but this is not indicated in the caption. It is not clear what the difference is. The styles of the drawings are different. It is necessary to arrange it in the same style. The numbers merge in the center of the graph. Better to fix it.

Answer: Thanks for your comments. Based on your comments, we revised figure 4.

Lines 349-350: Two seeds per pot were sown in PVC pipes. After seed germination, we selected one soybean plant for root analysis. Was the plant chosen at random or was there some kind of criterion?

Answer: First reason is that we wanted to select the plants that are in a uniform growth stage. As some soybean plants showed a difference in time for germination due to seed quality. To have different cultivars in similar growth stage, we selected one plant per PVC pipe.

Lines 353-354: Root samples were harvested when the seedlings reached the V2 growth stage (2 trifoliate leaves). Why was this growth stage chosen?

Answer: Before trefoil leaf emerges, soybean plants use energy sources like a carbohydrate for root development from an endosperm. However, after the trefoilate leaf emerges, the soybean plant uses a carbohydrate from the leaf which is generated by photosynthesis. Theoretically, the soybean plant produces energy sources from the leaf at the V1 growth stage. Most of the root is very well growing at the early growth stage. According to our experience, some soybean plant’s roots touch down the bottom of the PVC pipe at the V3 growth stage. If plant roots touch down the bottom of the PVC pipe, it is impossible to accurate data collection. So, we considered two factors for the collection of appropriate growth stage and we confirmed V2 was the best growth stage for root samples analysis.

Line 192: (Figure 4A). The biplot shows the loading of each variable (arrows) and scores of each Throughout the text, the letter designations of individual figures are lowercase, but here is an uppercase “A”.

Answer: It has been changed.

Lines 401-402 not referenced. Supplementary Materials: The following is available online at http: // ???, Table S1: List of soybean genotypes.

Answer: We were trying to provide some space to add the link for supplementary materials online, if accepted.

Reviewer 2 Report

The paper ´Investigation on root morphological traits using 2D-imaging among diverse soybeans (Glycine max L.)” written by Pooja Tripathi and co-authors deals with root morphology and architecture of plants. These root traits determine the productivity of plants in a variety of environments. The characteristics, such as root length, diameter, and surface area (SA), also support plant growth via water and nutrient absorption and provide anchorage to the plant and enhance the establishment of a symbiotic relationship with the microflora.
Unfortunately, from the exciting title, the paper is missing a scientific hypothesis and precise aim. The description is not enough for a scientific paper.

Therefore, an investigation on root system architecture is crucial for improving drought tolerance in many crops. Nevertheless, root morphological traits and root system architecture are essential in understanding osmotic and nutrient stress.
The conclusion is more a result summary. It can not be any conclusion without a scientific hypothesis.

Author Response

Reviewer 2
Comments: The paper ´Investigation on root morphological traits using 2D-imaging among diverse soybeans (Glycine max L.)” written by Pooja Tripathi and co-authors deals with root morphology and architecture of plants. These root traits determine the productivity of plants in a variety of environments. The characteristics, such as root length, diameter, and surface area (SA), also support plant growth via water and nutrient absorption and provide anchorage to the plant and enhance the establishment of a symbiotic relationship with the microflora.
Unfortunately, from the exciting title, the paper is missing a scientific hypothesis and precise aim. The description is not enough for a scientific paper.
Therefore, an investigation on root system architecture is crucial for improving drought tolerance in many crops. Nevertheless, root morphological traits and root system architecture are essential in understanding osmotic and nutrient stress.
The conclusion is more a result summary. It can not be any conclusion without a scientific hypothesis.

Answer: Thanks for your comments. Of course, I can accept your comment which was that “root system architecture is crucial for improving drought tolerance in many crops, so it is necessary needed for understanding osmotic and nutrient stress.” However, I can’t agree with your other comment “Unfortunately, from the exciting title, the paper is missing a scientific hypothesis and precise aim. The description is not enough for a scientific paper” due to below reason.
1) In soybean plant, root morphological traits did not identify among diverse soybean accession. For this reason, we need to preferentially ​figure out basic information of soybean root, like morphological traits. After that, further researches involved in root architectural traits are available. Baby can’t run without learning a baby step.